# Auranofin, identified by FDA-approved drug library screening, inhibits HBs antigen secretion via lysosomal damage

Akiyoshi Shimoda[1][⊙], Kazuhiro Murai[1][⊙], Hayato Hikita[1]*, Satoshi Minami[2], Takayuki Miyake[1], Shinji Kuriki[1], Emi Sometani[1], Jihyun Sung[1], Satoshi Shigeno[1], Yuichiro Higuchi[3], Kazuki Maesaka[1], Kumiko Shirai[1], Yuki Tahata[1], Yoshinobu Saito[1], Takahiro Kodama[1], Takeshi Takahashi[4], Hiroshi Suemizu[3], Tetsuo Takehara[1]

1 Department of Gastroenterology and Hepatology, The University of Osaka Graduate School of Medicine, Suita, Japan, 2 Department of Genetics, The University of Osaka Graduate School of Medicine, Suita, Japan, 3 Liver Engineering Laboratory, Department of Research for Humanized Model, Central Institute for Experimental Medicine and Life Science, Kawasaki, Japan, 4 Immunology Laboratory, Department of Medical Innovation, Central Institute for Experimental Medicine and Life Science, Kawasaki, Japan

⊙ These authors contributed equally to this work.

* hikita@gh.med.osaka-u.ac.jp

## Abstract

### Background and aim

Hepatitis B surface antigen (HBsAg) is associated with hepatocellular carcinoma risk and immune exhaustion and contributes to hepatitis B virus (HBV) persistence. Current anti-HBV treatments have a limited ability to reduce HBsAg levels. This study aimed to identify FDA-approved drugs capable of reducing HBsAg levels and explore the underlying mechanisms.

### Methods

A total of 1134 FDA-approved compounds were screened at 9.9 µM for 7 days using HepG2.2.15.7 cells, which contain an integrated HBV genome. HBsAg in the supernatant and cell viability were assessed. Compounds whose viability was reduced by >1 SD were excluded. Compounds whose HBsAg concentration decreased by >1.5 SD were selected and validated at 0.5 and 5 µM. We evaluated the HBsAg-reducing effect of the final candidate compounds using HBV-infected primary human hepatocytes (HepaSH cells) derived from chimeric TK-NOG mice and investigated the underlying mechanism responsible for the reduction in HBsAg.

### Results

After 126 cytotoxic compounds were excluded, the HBsAg levels of the 6 candidates decreased by >1.5 SD. Ethacridine and auranofin significantly reduced the level of HBsAg at both 5 and 0.5 µM. In HepaSH cells, only auranofin decreased the level

**Data availability statement:** All relevant data are within the manuscript and its Supporting Information files. The minimal data set underlying all figures and quantitative analyses, including numerical values used to generate graphs and data points extracted from images, has been provided as Supporting Information.

**Funding:** This study was funded by the Japan Agency for Medical Research and Development (AMED) under grant number JP25fk0310534 (K.M., H.H., T.K., and T.Taka.) and JP223fa627006 (Y.H., T.Taka., and H.S.), and Basis for Supporting Innovative Drug Discovery and Life Science Research (BINDS) (T.Take). Tetsuo Takehara received research grants from Gilead Sciences, Inc., GSK plc., and Bristol Myers Squibb. Hayato Hikita, Yuki Tahata, and Tetsuo Takehara received lecture fees from Gilead Sciences. None of the authors are employed by these commercial entities, and the authors declare that they have no other competing interests such as consultancy, patents, products in development, or marketed products. This does not alter our adherence to PLOS ONE policies on sharing data and materials.

**Competing interests:** Tetsuo Takehara received research grants from Gilead Sciences, Inc., GSK plc., and Bristol Myers Squibb. Hayato Hikita, Yuki Tahata, and Tetsuo Takehara received lecture fees from Gilead Sciences. This does not alter our adherence to PLOS ONE policies on sharing data and materials. There are no patents, products in development or marketed products associated with this research to declare.

of HBsAg. Auranofin did not affect the HBe antigen, HBV-DNA, or pregenomic RNA, nor did it reduce the level of intracellular HBsAg, as determined by Western blotting. Transmission electron microscopy revealed more vesicles in auranofin-treated HepaSH cells than in control cells. Immunofluorescence analysis of HepaSH cells treated with auranofin revealed increased Galectin-3 expression and colocalization of HBsAg with Galectin-3, which was consistent with lysosomal damage, compared with those of the untreated controls.

## Conclusion

Auranofin, an FDA-approved antirheumatic agent, reduces HBsAg secretion via lysosomal damage.

---

## Introduction

There are 254 million patients living with chronic hepatitis B, and approximately 1.1 million deaths occur due to cirrhosis and hepatocellular carcinoma (HCC) caused by hepatitis B infection [1]. The objectives of treatment are to suppress hepatitis activity, reduce liver fibrosis, prevent liver failure, inhibit carcinogenesis, and improve prognosis and quality of life (QOL). The current treatments for hepatitis B include interferon and nucleos(t)ide analogs (NUCs). Treatment with NUCs reduces serum HBV DNA levels, thereby suppressing chronic hepatitis [2–4]. Previous reports have shown that patients with low serum levels of HBV DNA have a lower risk of developing HCC than those with high serum levels of HBV DNA do [5–7]. However, the suppression of serum HBV DNA by NUCs does not completely prevent the development of HCC [8,9]. Moreover, compared with patients with low hepatitis B surface antigen (HBsAg) levels, patients with high HBsAg levels have a higher incidence of HCC, even when their HBV DNA levels are low and hepatitis B e antigen (HBeAg) is negative [10].

HBsAg is a protein transcribed and translated from the pre-S/S region of the HBV genome. There are three types of HBsAg: large HBsAg (LHBs), medium HBsAg (MHBs), and small HBsAg (SHBs). The pre-S1 region of LHBs plays important roles in adhesion to host cells, virion formation, and release from host cells [11,12]. According to previous reports, LHBs are carcinogenic, as transgenic mice overexpressing LHBs in the liver develop severe liver damage and HCC [13]. Subviral particles, which lack the HBV genome, are released 100 million times more frequently than complete virions are from HBV-infected cells [14]. The enormous amount of HBsAg in these subviral particles inhibits the neutralization of HBs antibodies [15], affects several types of immune cells [16–18], and leads to immune exhaustion and persistent HBV infection [19–21]. On the basis of these findings, treatments aimed at reducing HBsAg levels may have the potential to suppress carcinogenesis and prevent immune exhaustion.

HBsAg seroclearance, also known as a functional cure, has been suggested both clinically and experimentally to improve the prognosis of chronic hepatitis B patients. However, current treatments for hepatitis B, including interferon and NUCs, are

limited in their ability to reduce HBsAg levels [22]. In recent years, drug repurposing has emerged as a promising strategy to accelerate the identification of novel treatments [23]. The drug development process can be greatly expedited by screening existing drugs whose safety profiles and pharmacokinetics are already well characterized. The U.S. Food and Drug Administration (FDA) drug library, which comprises a diverse collection of approved drugs, represents a valuable resource for repurposing efforts. The present study aims to identify potential new HBV therapeutic agents capable of reducing HBsAg levels by evaluating potential drug candidates within the FDA-approved drug library and analyzing their effectiveness against HBV.

## Materials and methods

### Cell culture

**HepG2 cells.** HepG2 cells were cultured in Dulbecco's modified Eagle's medium (DMEM) (D5796; Merck KGaA, Darmstadt, Germany) supplemented with 10% fetal bovine serum (Thermo Fisher Scientific, Waltham, MA, USA), 100 units/mL penicillin, 100 µg/mL streptomycin, and 0.25 µg/mL amphotericin B (Thermo Fisher Scientific, Waltham, MA, USA). HepG2 cells were seeded on 24-well type I collagen-coated plates (AGC Techno Glass Co., Ltd., Shizuoka, Japan) at $5 \times 10^4$ cells/well for plasmid transfection.

**HepG2.2.15.7 cells.** HepG2.2.15.7 cells, which were stably transfected with the HBV genome (genotype D), were cultured in DMEM (D5796; Merck KGaA, Darmstadt, Germany) supplemented with 10% fetal bovine serum (Thermo Fisher Scientific, Waltham, MA, USA), 100 units/mL penicillin, 100 µg/mL streptomycin, 0.25 µg/mL amphotericin B (Thermo Fisher Scientific, Waltham, MA, USA), and 250 µg/mL G418 disulfate aqueous solution (Nacalai Tesque, Kyoto, Japan) [24,25]. HepG2.2.15.7 cells were seeded on 96-well type I collagen-coated plates (AGC Techno Glass Co., Ltd., Shizuoka, Japan) at $1 \times 10^4$ cells/well for drug screening and validation, 24-well type I collagen-coated plates at $5 \times 10^4$ cells/well for RNA isolation, 12-well type I collagen-coated plates at $1 \times 10^5$ cells/well for DNA isolation and Western blotting, and 8-well type I collagen-coated culture slides (Corning Incorporated, Corning, NY, USA) at $1 \times 10^4$ cells/well for immunofluorescence staining.

**HepaSH cells.** HepaSH cells are primary human hepatocytes that have been isolated from humanized liver chimeric TK-NOG mice [26]. All the mice used in this study were maintained under controlled temperature and light conditions with free access to food and water at the Central Institute for Experimental Medicine and Life Science. The cells were cultured in DMEM (08456−36; Nacalai Tesque, Kyoto, Japan) supplemented with 10% fetal bovine serum (Thermo Fisher Scientific, Waltham, MA, USA), 100 units/mL penicillin, 100 µg/mL streptomycin, 0.25 µg/mL amphotericin B (Thermo Fisher Scientific, Waltham, MA, Germany), 32.2 µg/mL 2-phospho-L-ascorbic acid trisodium (Merck KGaA, Darmstadt, Germany), 0.02 µg/mL dexamethasone (Merck KGaA, Darmstadt, Germany), 0.005 µg/mL recombinant human epidermal growth factor (Merck KGaA, Darmstadt, Germany), 20 mM 4-(2-hydroxyethyl)-1-piperazineethanesulfonic acid (Nacalai Tesque, Kyoto, Japan), 25 µL/L human insulin solution (Merck KGaA, Darmstadt, Germany), 0.37% NaHCO3 (Nacalai Tesque, Kyoto, Japan), and 2% dimethyl sulfoxide (DMSO) (Nacalai Tesque, Kyoto, Japan). HepaSH cells were seeded on 24-well type I collagen-coated plates (AGC Techno Glass Co., Ltd., Shizuoka, Japan) at $4 \times 10^5$ cells/well for validation of candidate compounds and RNA isolation; 12-well type I collagen-coated plates at $8 \times 10^5$ cells/well for DNA isolation, Western blotting and samples for transmission electron microscopy; and 8-well type I collagen-coated culture slides (Corning Incorporated, Corning, NY, USA) at $2 \times 10^5$ cells/well for immunofluorescence staining.

All the cell lines were proven to be free from mycoplasma infection.

### HBV infection

Hep38.7-tet cells are cells in which the genome of HBV genotype D is integrated, and the secretion of HBV virions in these cells is controlled by tetracycline [24]. The cells were cultured in DMEM/F12 (11320−033; Merck KGaA, Darmstadt,

Germany) supplemented with 10% fetal bovine serum (Thermo Fisher Scientific, Waltham, MA, USA), 100 units/mL penicillin, 100 µg/mL streptomycin, 0.25 µg/mL amphotericin B (Thermo Fisher Scientific, Waltham, MA, USA), 400 µg/mL G418 disulfate aqueous solution (Nacalai Tesque, Kyoto, Japan), and 55 µL/L human insulin solution (Merck KGaA, Darmstadt, Germany). The supernatant of Hep38.7-tet cells was collected 4 times over a period of 7 days and filtered through a 0.45 µm PVDF filter (Merck KGaA, Darmstadt, Germany). The filtered supernatant was precipitated with 8% PEG8000, 0.1 M NaCl, and 0.01 M 4-(2-hydroxyethyl)-1-piperazineethanesulfonic acid (HEPES) (Nacalai Tesque, Kyoto, Japan) and incubated at 4°C for several days. After centrifugation, the precipitate was collected as the HBV inoculum. HepaSH cells were incubated at 37°C under 5% $CO_2$ for 3 days after seeding and then incubated for 1 day with HBV inoculum (1000 GEq/cell) and medium supplemented with 4% PEG8000. The medium was changed to fresh medium every 5 days.

### Drug screening and validation

The FDA-approved drug library was kindly provided by the Center for Supporting Drug Discovery and Life Science Research, The University of Osaka Graduate School of Pharmaceutical Sciences. A library of 1134 FDA-approved drugs was screened on HepG2.2.15.7 cells at a concentration of 9.9 µM for 7 days. After 7 days of incubation, the levels of HBsAg in the supernatant were measured, and cell viability was assessed using a WST-8 (Nacalai Tesque, Kyoto, Japan). Compounds that reduced HepG2.2.15.7 cell viability by more than 1 standard deviation (SD) were excluded. Compounds whose level of HBsAg in the supernatant decreased by more than 1.5 SD were selected as candidate compounds. These candidates were then validated at concentrations of 5 µM and 0.5 µM using the same time course. Compounds that significantly reduced the amount of HBsAg in the supernatant at both concentrations were selected as the final candidates. To elucidate the mechanism of HBsAg reduction, HepG2.2.15.7 cells and HBV-inoculated HepaSH cells were treated with the final candidates.

### Compounds

Auranofin (S4307) and ethacridine lactate monohydrate (S4196) were purchased from Selleck Chemicals (Houston, TX, USA).

### Measurement of HBsAg, HBeAg, HBV DNA, and albumin levels

HBsAg levels were measured using a chemiluminescent enzyme immunoassay (CLEIA; LUMIPULSE Presto HBsAg-HQ; Fujirebio, Tokyo, Japan). Hepatitis B e antigen (HBeAg) levels were measured using a CLEIA (LUMIPULSE Presto HBeAg; Fujirebio, Tokyo, Japan). The samples were diluted 50-fold. HBV DNA levels were measured using the cobas 6800/8800 system (Roche Diagnostics, Rotkreuz, Switzerland). The samples were diluted 100-fold. Albumin levels were measured using a Human Albumin ELISA Quantitation Kit (Bethyl Laboratories, Inc.; Montgomery, TX, USA).

### Western blotting

The Western blotting procedure was based on a previous report [27]. The membrane was blocked with Tris-buffered saline, polysorbate 20, and 5 g/mL skim milk. The primary antibodies used were against JNK (#9258), eIF2 α (#5324), IRE1 α (#3294), PERK (#3192), CHOP (#2895) (Cell Signaling Technology, Danvers, MA, USA), pIRE1 α (NB100–2323; Novus Biotechnology, Centennial, CO, USA), beta actin (A5316; Merck KGaA, Darmstadt, Germany), and HBsAg (Bs-1557G Bioss (Woburn, MA, USA) for the HepG2.2.15.7 cell samples and NB100–62652 (Novus Biologicals, Centennial, CO, USA) for the HepaSH cell samples. These primary antibodies were diluted 1:1000–1:10000 with Can Get Signal Immunoreaction Enhancer Solution 1 (Toyobo, Osaka, Japan). The secondary antibodies used were mouse (NA931), rabbit (NA934), rat (NA935) (Cytiva, Marlborough, MA, USA), and goat (PO449; Agilent Technologies, Santa Clara, CA, USA), and they were diluted 1:3000–1:5000 with blocking buffer.

## Real-time quantitative polymerase chain reaction (RT–qPCR) and droplet digital PCR (ddPCR)

RNA was isolated from samples using the RNeasy Mini Kit (Qiagen, Venlo, Netherlands) according to the manufacturer's instructions, with DNase I (RNase-Free DNase Set, Qiagen, Venlo, Netherlands) added to remove genomic DNA during isolation. The isolated RNA was then reverse transcribed using ReverTra Ace qPCR RT Master Mix (Toyobo, Osaka, Japan), and RT–qPCR was performed with a QuantStudio 6 Flex Standard Real-Time System (Thermo Fisher Scientific, Waltham, MA, USA) using TaqMan Gene Expression Assays (Thermo Fisher Scientific, Waltham, MA, USA). The following primer and probe sets were used for analysis: pgRNA, with forward primer 5'-TGTCCTACTGTTCAAGCCTCCAA-3', probe 5'-CATGGACATCGACCC-3', and reverse primer 5'-GAGAGTAACTCCACAGTAGCTCCAA-3', LGALS3 (Galectin-3) (Hs03680062_m1; Thermo Fisher Scientific, Waltham, MA, USA), TXNRD1 (Hs00917067_m1; Thermo Fisher Scientific, Waltham, MA, USA), TXNRD2 (Hs00272352_m1; Thermo Fisher Scientific, Waltham, MA, USA) and human β-actin (Hs99999903_m1; Thermo Fisher Scientific, Waltham, MA, USA) as the internal controls.

DNA was isolated from samples using a QIAamp DNA Mini Kit (Qiagen Venlo, Netherlands) according to the manufacturer's instructions. The isolated DNA was then analyzed by RT–qPCR with a QuantStudio 6 Flex Standard Real-Time System (Thermo Fisher Scientific, Waltham, MA, USA) using TaqMan Gene Expression Assays (Thermo Fisher Scientific, Waltham, MA, USA).

The level of cccDNA was analyzed by ddPCR using the QuantStudio Absolute Q digital PCR system (Thermo Fisher Scientific Waltham, MA, USA). Before analysis, the isolated DNA was digested with Plasmid-Safe ATP-Dependent DNase (Biosearch Technologies, Hoddesdon, UK) [28].

The following primer and probe sets were used for analysis: HBV-DNA, with forward primer 5'-ACATCAGGATTCCTAGGACCC-3', probe 5'- CAGAGTCTAGACTCGTGGTGGACTTC-3', and reverse primer 5'- GGTGAGTGATTGGAGGTTGG-3'; cccDNA, with forward primer 5'-CGTCTGTGCCTTCTCATCTGC-3', probe 5'-CTGTAGGCATAAATTGGT-3' and reverse primer 5'-GCACAGCTTGGAGGCTTGAA-3'; and RNase P (440332; Thermo Fisher Scientific, Waltham, MA, USA) as the internal control.

## Plasmid DNA transfection

pHBV-Luc (Addgene plasmid # 71414), pHBV-S1-Luc (Addgene plasmid # 71416), and pHBV-S2-Luc (Addgene plasmid # 71417) plasmids were purchased from Addgene (Watertown, MA, USA) and provided by Wang-Shick Ryu [29,30]. pHBV-Luc is a plasmid designed to evaluate core promoter activity via luciferase luminescence, with the luciferase gene placed downstream of the HBV Enhancer I (EnhI), Enhancer II (EnhII), and basal core promoter (BCP) regions. Similarly, pHBV-S1-Luc and pHBV-S2-Luc are plasmids designed to assess S1 and S2 promoter activity, respectively, through luciferase luminescence, with the luciferase gene positioned downstream of the S1 and S2 promoter regions. Plasmids were transfected into HepG2 cells using Lipofectamine 2000 (Thermo Fisher Scientific, Waltham, MA, USA), and the cells were incubated for three days with or without auranofin or with IFN (500 IU/mL) as a positive control.

TXNRD1 (VB9000006−8933prq), TXNRD2 (VB9000010−2316vwn), and EGFP control vector (VB10000−9288rhy) plasmids were purchased from Vector Builder (Chicago, IL, USA). The plasmids were transfected into HepG2.2.15.7 cells using Lipofectamine 2000 (Thermo Fisher Scientific, Waltham, MA, USA), and the cells were incubated for three days.

## Promoter activity evaluation

Promoter activity was measured using the Dual-Luciferase Reporter Assay System (Promega, WI, USA) according to the manufacturer's instructions.

## Proteasome activity

HepG2.2.15.7 cells were incubated with auranofin for 3 days, after which proteasome activity was assessed. HepG2.2.15.7 cells were treated with 1 μM mitoxantrone (MTX) as a positive control. Proteasome activity was measured

using a CycLex Proteasome Enrichment and Activity Assay Kit (MBL, Tokyo, Japan). In accordance with the manufacturer's instructions, 20S proteasomes were isolated from cell lysates using hHR23B ubiquitin-like domain resin (UbL-Resin). The cell lysate of each sample was treated with/without UbL-Resin. Suc-Leu-Leu-Val-Tyr-7-amino-4-methylcoumarin (Suc-LLVY-AMC) was added, and fluorescence was measured with a Varioskan LUX multimode microplate reader (Thermo Fisher Scientific, Waltham, MA, USA) for 2 hours at 5-minute intervals.

## Transmission electron microscopy

HepaSH cells were either treated with 1 μM auranofin for 5 days or left untreated. After treatment, the cells were washed 3 times with PBS and fixed with 2.5% glutaraldehyde in 0.1 M phosphate buffer (Nacalai Tesque, Kyoto, Japan) for 2 hours on ice. The fixed samples were then embedded in epoxy resin, sectioned, and stained for electron microscopy. The samples were examined using an HT7800 transmission electron microscope (HITACHI, Tokyo, Japan).

## Immunofluorescence staining

HepaSH cells were either treated with 1 μM auranofin for 5 days or left untreated. The samples were washed with PBS three times and fixed for 10 minutes at room temperature with 4% paraformaldehyde phosphate buffer solution (Nacalai Tesque, Kyoto, Japan). After another three washes with PBS, the samples were permeabilized for 10 minutes at room temperature with a solution containing 0.2% Triton X-100 (Nacalai Tesque, Kyoto, Japan), 0.2% bovine serum albumin (BSA) (Nacalai Tesque, Kyoto, Japan), and 0.04% sodium azide (Nacalai Tesque, Kyoto, Japan) in PBS. The samples were subsequently washed with PBS three more times and blocked with 0.2% BSA in PBS at room temperature for 30 minutes, after which the buffer was discarded. The samples were then incubated with primary antibodies for one hour at room temperature, washed three times with PBS, and then incubated with secondary antibodies for one hour at room temperature in the dark. The samples were subsequently washed three times with PBS, mounted with DAPI Fluoromount-G (SouthernBiotech, Bermingham, AL, USA), and examined using an FV4000 laser scanning confocal microscope (EVIDENT, Tokyo, Japan). The primary antibodies used were Galectin-3 (sc-23938; Santa Cruz Biotechnology, Dallas, TX, USA) and HBsAg (NB100–62652 Novus Biologicals, Centennial, CO, USA). These primary antibodies were diluted to a concentration of 1:500 with Can Get Signal Immunostain Immunoreaction Enhancer Solution B (Toyobo, Osaka, Japan). The secondary antibodies used were goat anti-rat (Alexa Fluor 488; Thermo Fisher Scientific, Waltham, MA, USA) and goat anti-rabbit (Alexa Fluor 594; Thermo Fisher Scientific, Waltham, MA, USA), and they were diluted to a concentration of 1:500 with Can Get Signal Immunoreaction Enhancer Solution B (Toyobo, Osaka, Japan).

## Statistical analysis

Statistical analyses were performed with JMP version 17.1.0 (SAS Institute, Cary, NC, USA). The Dunnett test was used to compare multiple groups. A t test was used to compare two values. Correlations were assessed using Pearson's correlation coefficient. A P value <0.05 was considered significant. The minimal data set underlying all figures and quantitative analyses, including the numerical values used to generate graphs and data points extracted from images, is provided in the Supporting Information (S1 Data).

## Ethics statement

This study did not involve human participants, human data, or human tissue. Therefore, approval by an institutional review board or ethics committee and informed consent were not needed. All experimental procedures involving animals were conducted in accordance with the institutional guidelines for animal care and were approved by the Animal Care Committee of the Central Institute for Experimental Medicine and Life Science.

## Results

### An FDA-approved drug screening to reduce HBsAg in the supernatant identified auranofin

A total of 126 compounds were excluded because of reduced cell viability below 1 SD (Fig 1A). Among the remaining 1008 compounds, 6 compounds were identified as candidate compounds whose HBsAg levels in the supernatant decreased by more than 1.5 SD (Table 1, Fig 1B and 1C). HepG2.2.15.7 cells were treated with each compound at concentrations of 0, 0.5, and 5 µM, and the HBsAg concentration and cell viability were determined. No decrease in cell viability was observed at any concentration. However, only ethacridine lactate monohydrate and auranofin significantly decreased the level of HBsAg in the supernatant at concentrations of 0.5 and 5 µM (Fig 1D). In HepaSH cells (primary human hepatocytes that were isolated from humanized liver chimeric TK-NOG mice), although ethacridine lactate monohydrate did not decrease the level of HBsAg in the culture supernatant, auranofin did decrease the level of HBsAg in the supernatant. Notably, neither ethacridine lactate monohydrate nor auranofin decreased cell viability (Fig 1E and 1F).

### Auranofin reduced only the supernatant HBsAg levels

HepG2.2.15.7 cells were treated with various concentrations of auranofin, and the rate of decrease was consistent within a specific range. The $EC_{50}$ of auranofin in HepG2.2.15.7 cells was 0.375 µM (Fig 2A). On the basis of this range, we selected 1 µM auranofin for the experiments. Treatment of HepG2.2.15.7 cells with auranofin reduced the level of HBsAg in the supernatant but did not affect the levels of HBeAg, HBV DNA in the supernatant, or intracellular pregenomic RNA (pgRNA) or HBV DNA (Fig 2B). Furthermore, auranofin treatment did not alter intracellular HBs protein expression in HepG2.2.15.7 cells (Fig 2C). Treatment of HBV-infected HepaSH cells and HepG2.2.15.7 cells with auranofin reduced the level of HBsAg in the supernatant but did not affect the levels of HBeAg or HBV DNA in the supernatant, intracellular pgRNA, HBV DNA, covalently closed circular DNA (cccDNA) (Fig 2D), or HBs protein expression (Fig 2E). In HepG2.2.15.7 cells and HBV-infected HepaSH cells, auranofin treatment did not reduce the albumin levels in the culture supernatant (S1 Fig). The $EC_{50}$ of auranofin in HepaSH cells was 6.236 µM (S2 Fig).

### Auranofin did not affect proteasome activity, endoplasmic reticulum stress, or HBV promoter activity

We investigated the mechanisms underlying the reduction in extracellular HBsAg in HepG2.2.15.7 cells treated with auranofin. First, HBV promoter activity was assessed using a promoter assay to evaluate transcriptome activity, but auranofin treatment did not affect promoter activity (Fig 3A). It has been previously reported that the induction of endoplasmic reticulum (ER) stress in hepatocytes suppresses HBsAg secretion [31]. Therefore, to determine whether auranofin induces ER stress in hepatocytes, we examined changes in the expression of ER stress-related proteins with or without auranofin treatment. Auranofin did not significantly alter the expression of these genes (Fig 3B). Finally, proteasome activity was analyzed to determine whether HBsAg degradation was enhanced, but no alterations were detected (Fig 3C).

**Table 1. Candidate compounds with HBsAg levels in the supernatant decreased by more than 1.5 SD.**

|   | Drug name | HBs/Ctrl | WST/Ctrl |
|---|---|---|---|
| 1 | Ethacridine lactate monohydrate | 0.308 | 1.09 |
| 2 | Imatinib (STI571) | 0.314 | 0.909 |
| 3 | Auranofin | 0.325 | 0.838 |
| 4 | Otilonium Bromide | 0.332 | 0.757 |
| 5 | Carbadox | 0.360 | 0.825 |
| 6 | Flubendazole | 0.388 | 0.699 |

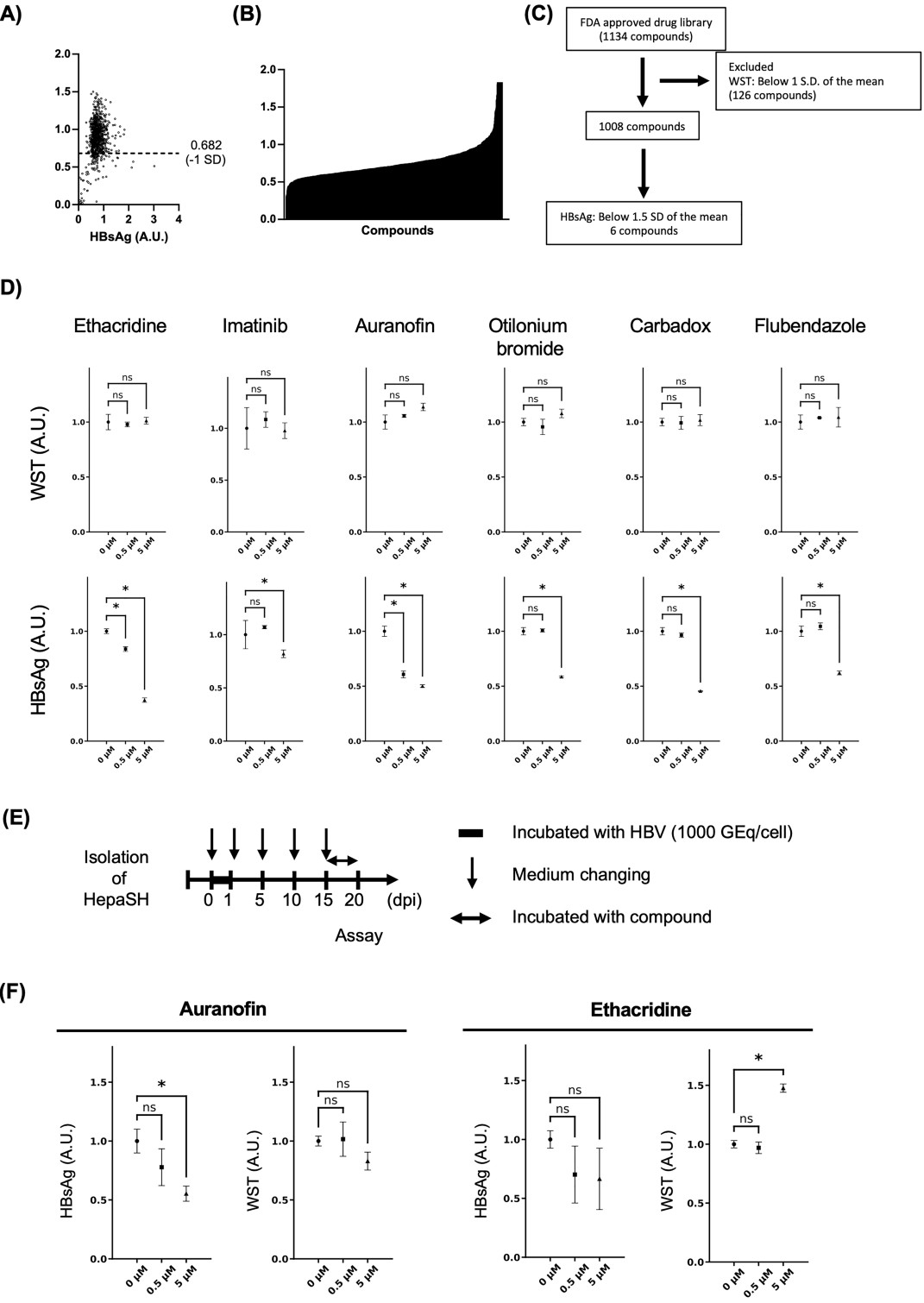

**Fig 1. FDA-Approved Drug Library Screening. (A)** Cell viability and HBsAg levels after treatment with the respective compounds. **(B)** HBsAg was treated with the remaining 1008 compounds. **(C)** Six compounds whose HBsAg levels decreased by more than 1.5 SD were identified as candidate compounds. **(D)** Cell viability and HBsAg levels after treatment with 6 compounds at concentrations of 0, 0.5, and 5 μM. **(E)** HBV-inoculated HepaSH cells (1000 GEq/cell) were treated with ethacridine or auranofin. **(F)** Viability and HBsAg levels of HBV-inoculated HepaSH cells treated with auranofin or ethacridine (n = 3. *: p < 0.05).

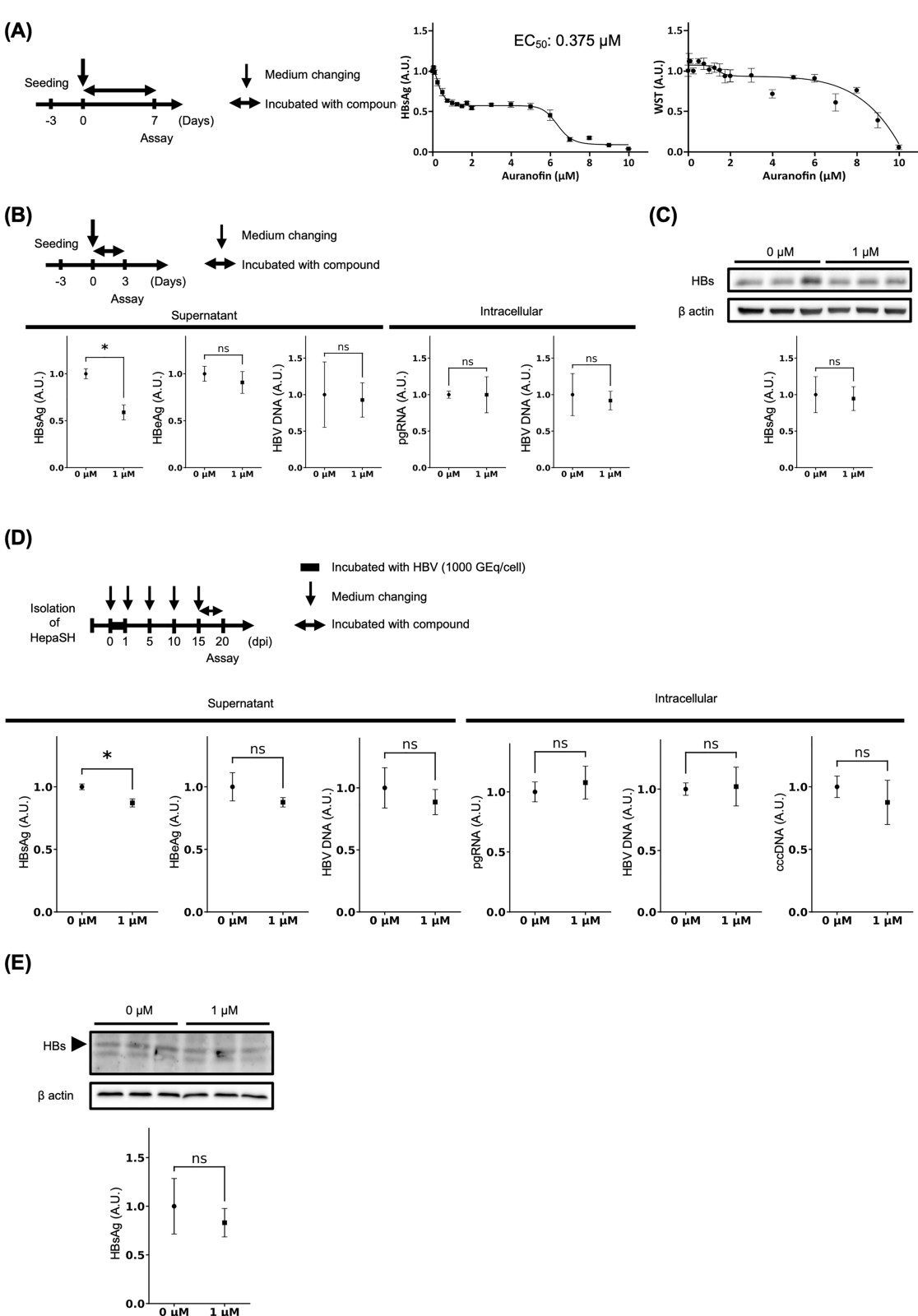

**Fig 2. Auranofin Specifically Reduced HBsAg in the Supernatant of HepG2.2.15.7 and HBV-Inoculated HepaSH Cells.** HepG2.2.15.7 cells (n = 3) and HBV-inoculated HepaSH cells (1000 GEq/cell, n = 4) were treated with 1 μM auranofin. **(A)** HBsAg levels in the supernatant and cell viability

of HepG2.2.15.7 cells treated with various concentrations of auranofin. **(B)** HBsAg, HBeAg, and HBV DNA levels in the supernatant and intracellular pgRNA and HBV DNA levels in HepG2.2.15.7 cells treated with or without auranofin. (*: $p < 0.05$). **(C)** Western blotting and band quantification of intracellular HBs proteins in HepG2.2.15.7 cells treated with or without auranofin. **(D)** HBsAg, HBeAg, and HBV DNA levels in the supernatant and intracellular pgRNA, HBV DNA and cccDNA levels in HBV-inoculated HepaSH cells treated with or without auranofin. (*: $p < 0.05$). **(E)** Western blotting and band quantification of intracellular HBs proteins in HBV-inoculated HepaSH cells treated with or without auranofin.

## Auranofin reduced the extracellular secretion of HBsAg by inducing lysosomal damage

To examine the structural changes induced by auranofin in hepatocytes, HepaSH cells were observed by transmission electron microscopy (TEM) after treatment with auranofin. TEM revealed more vesicles in auranofin-treated HBV-infected HepaSH cells than in untreated HBV-infected HepaSH cells. These vesicles were not found in noninfected HepaSH cells (Fig 4A and 4B). It has been previously reported that auranofin induces lysosomal damage through the inhibition of thioredoxin reductase (TrxR) [32,33]. Considering that the number of vesicles observed by TEM, which increased in HBV-infected HepaSH cells after auranofin treatment, might indicate damaged lysosomes, we evaluated the colocalization of damaged lysosomes and HBsAg by immunofluorescence staining. We evaluated Galectin-3 as a specific protein marker of damaged lysosomes [34]. Immunofluorescence staining revealed that auranofin treatment increased Galectin-3 expression in HBV-infected HepaSH cells (Fig 4C). Treatment of HepG2.2.15.7 cells with auranofin led to a dose-dependent increase in Galectin-3 mRNA expression. In addition, Galectin-3 mRNA expression levels were negatively correlated with HBsAg levels in the culture supernatant (S3 Fig). In the absence of auranofin, no colocalization of HBsAg and Galectin-3 was observed, whereas auranofin treatment induced the colocalization of HBsAg and Galectin-3 in HBV-infected HepaSH cells (Fig 4C). In HepG2.2.15.7 cells, the thioredoxin reductases TXNRD1 and TXNRD2 were overexpressed. Compared with the control group, the group in which TXNRD1 was overexpressed did not have altered HBsAg levels in the culture supernatant, whereas the group in which TXNRD2 was overexpressed had increased extracellular HBsAg levels (S4 Fig). Taken together, these findings indicate that auranofin-induced lysosomal damage leads to Galectin-3 accumulation and HBsAg adsorption to damaged lysosomes, resulting in reduced extracellular HBsAg secretion.

## Discussion

In the present study, we identified auranofin through an FDA-approved drug library screening as a compound that reduces extracellular HBsAg levels. While auranofin reduced extracellular HBsAg levels, it did not decrease extracellular HBeAg, HBV DNA, or intracellular pgRNA, cccDNA, or HBsAg levels. Auranofin did not affect proteasome activity, endoplasmic reticulum stress, or HBV promoter activity. We revealed that auranofin induces Galectin-3 expression through lysosomal damage, and Galectin-3 colocalizes with HBsAg, thereby inhibiting its extracellular secretion. This is the first study to identify a drug with a mechanism for inhibiting HBsAg secretion through an FDA-approved drug library screening.

The spontaneous loss of HBsAg in chronic HBV patients is rare, occurring at a rate of 0.5–1% per year. HBsAg loss with current treatments ranges from 3–7% with pegylated IFN and 0–3% with NUCs [22]. Research and development of new therapies for treating HBV are ongoing. Antisense oligonucleotides (ASOs) targeting HBV messenger RNA are expected to reduce HBsAg levels; however, the results from a phase 2b trial demonstrated that HBsAg seroclearance was achieved in only approximately 9–10% of participants treated with ASOs [35]. It is difficult to achieve HBsAg clearance with a single agent, including current therapies and those under clinical investigation. Therefore, novel treatments that can efficiently reduce HBsAg levels are urgently needed.

Auranofin, a gold-containing compound traditionally used as an orally administered antirheumatic agent, has promising multifunctional effects, including anti-inflammatory, anticancer, and anti-infection effects, although the mechanisms

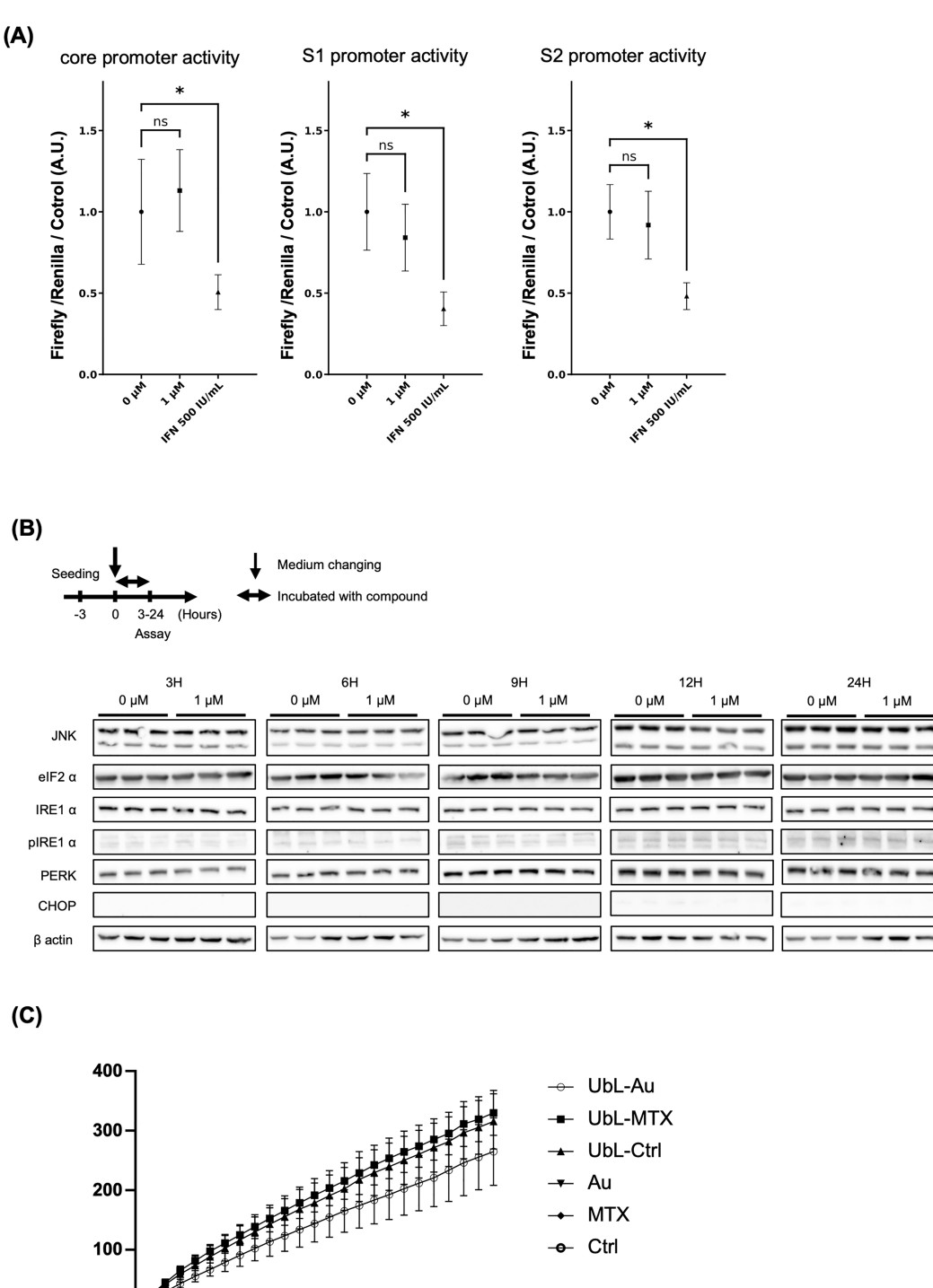

**Fig 3. Auranofin did not affect proteasome activity, endoplasmic reticulum stress, or HBV promoter activity. (A)** HBV promoter activity with or without auranofin or with IFN (500 IU/mL) in HepG2.2.15.7 cells. (n = 3, *: p < 0.05). **(B)** Western blotting of ER stress-related proteins from 3 to 24 hours after auranofin treatment in HepG2.2.15.7 cells. **(C)** Proteasome activity in HepG2.2.15.7 cells was assessed by isolation of 20S proteasomes from cell lysates using an hHR23B ubiquitin-like domain resin (UbL-resin). The cell lysate of each sample was treated with/without UbL-Resin. (UbL-: samples prepared with UbL-Resin; Au: auranofin; MTX: mitoxantrone (positive control); Ctrl: control).

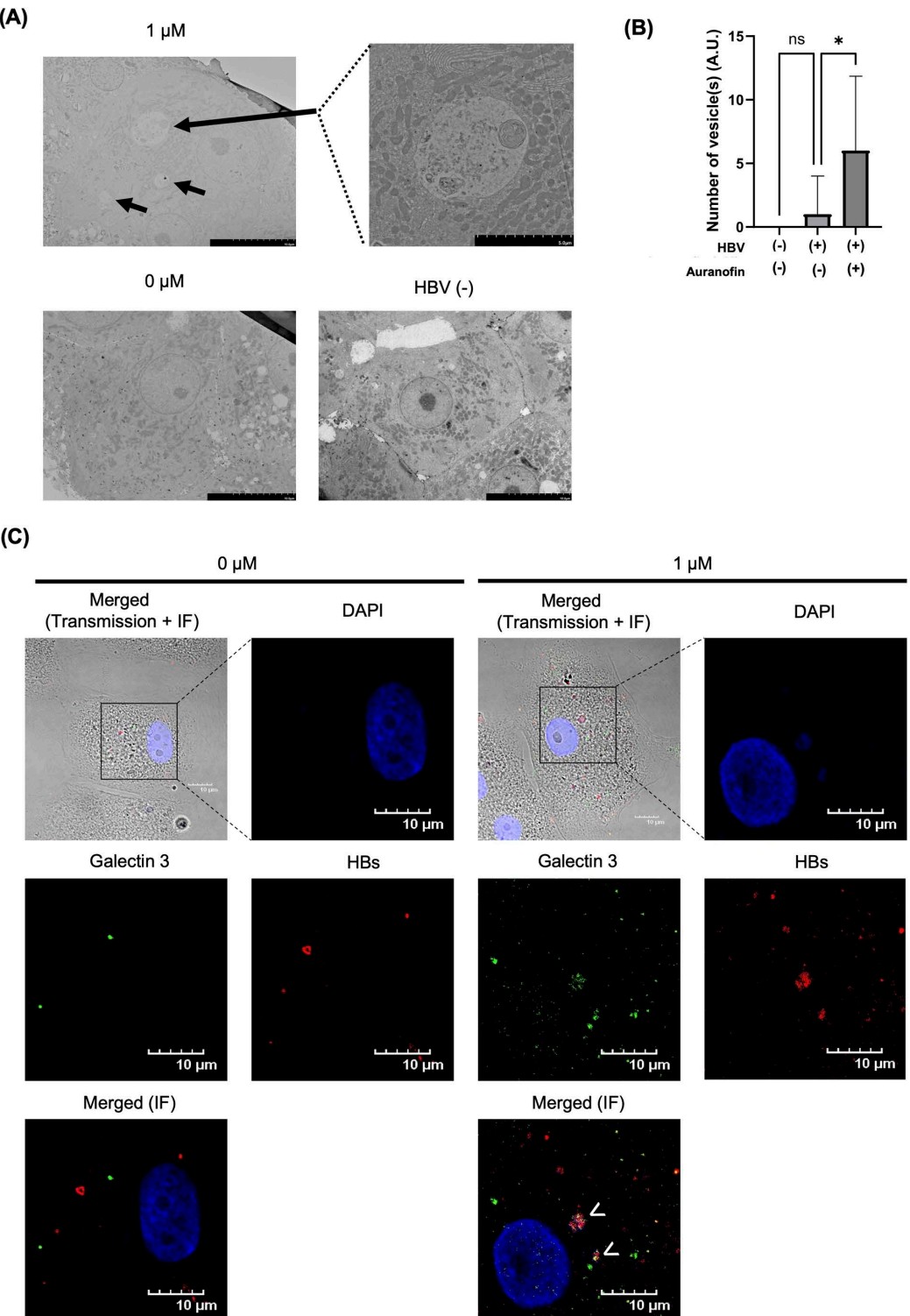

**Fig 4. Auranofin Reduced the Extracellular Secretion of HBsAg by Inducing Lysosomal Damage. (A)** Transmission electron microscopy images of HBV-infected HepaSH cells treated with or without auranofin and noninfected HepaSH cells. **(B)** Number of vesicles in the cytoplasm of HBV-infected HepaSH cells treated with or without auranofin and noninfected HepaSH cells observed by electron microscopy. **(C)** Immunofluorescence staining of HBV-infected HepaSH cells treated with or without auranofin.

underlying these properties remain unclear [36]. Auranofin has been reported to have anti-inflammatory effects by inhibiting degranulation and superoxide production in specific leukocytes [37], suppressing antigen presentation in antigen-presenting cells, and inactivating T-cell responses [38]. It also inhibits NF-κB signaling [39] and decreases complement C3 levels via the inhibition of JAK/STAT3 signaling [40]. The anticancer properties of auranofin include proteasome inhibition [41,42], caspase activation [42], inhibition of thioredoxin reductase (TrxR) activity [43,44], and induction of ER stress and the UPR [41,43] in several cell lines. As an antiviral agent, auranofin reduces the reservoir of HIV-infected memory T cells [45], inhibits the oxidative folding pathway, causing the misfolding of chikungunya virus (CHIKV) proteins, and inhibits CHIKV replication [46]. Additionally, auranofin has been shown to inhibit SARS-CoV-2 replication and reduce intracellular cytokine mRNA levels [47]. There have been no reports on auranofin and HBV replication.

Galectin-3 is one of the lectins encoded by the LGALS3 gene and is located in the cytoplasm, nucleus, and membrane. One of the functions of galectin3 is binding to luminal glycans upon lysosomal damage [48]. Galectin-3 within damaged lysosomes is known to bind various proteins and thereby induce physiological processes, such as the recruitment of ALIX to activate endosomal sorting complexes required for transport (ESCRT)-mediated repair or binding to TRIM16 to initiate autophagy [49]. In the present study, auranofin-induced lysosomal damage led to Galectin-3 accumulation and HBsAg adsorption to damaged lysosomes, resulting in reduced extracellular HBsAg secretion. However, we did not directly demonstrate a causal relationship between lysosomal injury and impaired HBsAg secretion. To clarify this mechanistic link more conclusively, further studies aimed at directly modulating lysosomal integrity are needed.

Several studies have focused on screening compound libraries for agents that reduce HBsAg levels. Tetrahydro-tetrazolo-pyrimidine was identified through screening with the IHVR small-molecule collection and HepG2.2.15 cells. This compound inhibits the secretion of HBsAg without affecting viral replication, although an accumulation of intracellular HBsAg is observed [50]. In the present study, auranofin reduced the extracellular secretion of HBsAg without causing its intracellular accumulation. Further investigations are needed to elucidate the mechanism through which auranofin prevents the intracellular accumulation of HBsAg. Another compound, 6-aminonicotinamide, was identified through screening with a commercial-based library of small-molecule compounds and HepAD38 cells. This compound inhibited HBV SI, SII, and core promoter activity by decreasing the transcription factor peroxisome proliferator-activated receptor α (PPARα), resulting in a decrease in HBV RNA transcription [51]. However, none of these compounds has been approved by the FDA.

In conclusion, the compound auranofin, identified from an FDA-approved drug screening, decreases HBsAg secretion as a result of lysosomal damage.

## Supporting information

**S1 Fig. Auranofin treatment did not reduce the albumin levels in the culture supernatant.** (A) Albumin levels in the supernatant of HepG2.2.15.7 cells treated with or without auranofin. (B) Albumin levels in the supernatant of HBV-infected HepaSH cells treated with or without auranofin.
(TIFF)

**S2 Fig. HBsAg levels in the supernatant of HepaSH cells treated with various concentrations of auranofin.**
(TIFF)

**S3 Fig. Treatment of HepG2.2.15.7 cells with auranofin led to a dose-dependent increase in Galectin-3 mRNA expression, and Galectin-3 mRNA expression levels were negatively correlated with HBsAg levels in the culture supernatant.** (A) HBsAg levels in the supernatant and mRNA expression levels of Galectin-3 in HepG2.2.15.7 cells treated with various concentrations of auranofin. (B) Correlations between HBsAg levels in the supernatant and Galectin-3 mRNA expression levels in HepG2.2.15.7 cells.
(TIFF)

**S4 Fig. TXNRD2 overexpression resulted in an increase in extracellular HBsAg levels.** HBsAg levels in the supernatant, cell viability, and TXNRD1/TXNRD2 mRNA expression levels in HepG2.2.15.7 cells with or without TXNRD1 or TXNRD2 overexpression. (*: $p < 0.05$).
(TIFF)

**S1 Data. The minimal data set underlying all figures and quantitative analyses, including numerical values used to generate graphs and data points extracted from images.**
(XLSX)

## Acknowledgments

The embedding, sectioning, and staining of the TEM samples were supported by Tomoaki Mizuno, Center for Medical Research and Education, The University of Osaka Graduate School of Medicine.

## Author contributions

**Conceptualization:** Akiyoshi Shimoda, Kazuhiro Murai, Hayato Hikita.

**Data curation:** Akiyoshi Shimoda, Kazuhiro Murai, Hayato Hikita, Satoshi Minami, Takayuki Miyake, Shinji Kuriki, Emi Sometani, Jihyun Sung, Satoshi Shigeno, Yuichiro Higuchi, Kazuki Maesaka, Kumiko Shirai, Yuki Tahata, Yoshinobu Saito, Takahiro Kodama, Takeshi Takahashi, Hiroshi Suemizu.

**Formal analysis:** Akiyoshi Shimoda, Kazuhiro Murai, Hayato Hikita, Satoshi Minami, Takayuki Miyake, Shinji Kuriki, Emi Sometani, Jihyun Sung, Satoshi Shigeno, Yuichiro Higuchi, Kazuki Maesaka, Kumiko Shirai, Yuki Tahata, Yoshinobu Saito, Takahiro Kodama, Takeshi Takahashi, Hiroshi Suemizu.

**Project administration:** Tetsuo Takehara.

**Resources:** Yuichiro Higuchi, Takeshi Takahashi, Hiroshi Suemizu.

**Supervision:** Tetsuo Takehara.

**Writing – original draft:** Akiyoshi Shimoda, Kazuhiro Murai.

**Writing – review & editing:** Tetsuo Takehara.

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
