## [Decision Letter · Decision Letter 0]

24 Oct 2025

Dear Dr. Hikita,

Thank you for submitting your manuscript to PLOS ONE. After careful consideration, we feel that it has merit but does not fully meet PLOS ONE’s publication criteria as it currently stands. Therefore, we invite you to submit a revised version of the manuscript that addresses the points raised during the review process.

Authors should revise their manuscript accordingly. Thank you very much for hundling their nice manuscript.

We look forward to receiving your revised manuscript.

Kind regards,

Tatsuo Kanda, M.D.; Ph.D.

Academic Editor

PLOS ONE

Journal Requirements:

This study was funded by the Japan Agency for Medical Research and Development (AMED) under grant numbers JP25fk0310534 (K.M., H.H., T.K., and T. Taka.) and JP223fa627006 (Y.H., T. Taka., and H.S.), and by the Basis for Supporting Innovative Drug Discovery and Life Science Research (BINDS).

4. Thank you for stating the following in the Competing Interests/Financial Disclosure section:

Tetsuo Takehara received research grants from Gilead Sciences, Inc, GSK plc. and Bristol Myers Squibb. Hayato Hikita, Yuki Tahata, and Tetsuo Takehara received lecture fees from Gilead Sciences.

We note that one or more of the authors are employed by commercial companies: Gilead Sciences, Inc, GSK plc. and Bristol Myers Squibb

6. We note that your Data Availability Statement is currently as follows: All relevant data are within the manuscript and its Supporting Information files.

Reviewers' comments:

Reviewer's Responses to Questions

**Comments to the Author**

1. Is the manuscript technically sound, and do the data support the conclusions?

Reviewer #1: Partly

Reviewer #2: Yes

2. Has the statistical analysis been performed appropriately and rigorously?

Reviewer #1: Yes

Reviewer #2: Yes

3. Have the authors made all data underlying the findings in their manuscript fully available?

Reviewer #1: Yes

Reviewer #2: Yes

4. Is the manuscript presented in an intelligible fashion and written in standard English?

Reviewer #1: Yes

Reviewer #2: Yes

Reviewer #1: The authors focused on auranofin as a potential novel therapeutic agent for HBV infection.

Auranofin failed to decrease intracellular HBsAg or HBV RNA. The authors showed that auranofin only reduced extracellular HBsAg release and increases the number of intracellular Galectin-3–positive vesicles in HBV-infected cells. Based on these findings, the authors concluded—and reflected in the title—that lysosomal damage is responsible for the reduced extracellular release of HBsAg. Since the major novelty of this study lies in the finding that auranofin is associated with reduced HBsAg release, it would be essential to elucidate the molecular mechanism underlying this effect. Without mechanistic evidence linking lysosomal damage directly to impaired HBsAg secretion, publication of this work would be difficult to justify.

#1. While the increase in Galectin-3 and damaged lysosomes after auranofin treatment in HBV-infected HepaSH cells suggests lysosomal injury, this observation alone does not directly demonstrate that lysosomal damage contributes to the reduced extracellular release of HBsAg.The authors could perform rescue experiments using antioxidants or lysosomal membrane–stabilizing interventions to determine whether preventing lysosomal damage restores HBsAg secretion under Auranofin treatment.

#2. The correlation between the degree of Galectin-3 expression or the number of Galectin-3–positive vesicles with the amount of extracellular HBsAg would strengthen the conclusion. Although no clear dose-dependency of auranofin was observed, this point is somewhat concerning, as it raises questions about the reproducibility and mechanism of the observed effects. A dose–response or time-course analysis showing that lysosomal damage (as indicated by Galectin-3 puncta) quantitatively associates with decreased HBsAg release would provide more convincing mechanistic evidence.

Reviewer #2: This is a well-designed repositioning study that identifies auranofin as a reducer of HBsAg via inhibition of extracellular release. The experimental design and mechanistic dissection are coherent, and the findings are virologically insightful. I recommend publication after the following minor points are addressed:

1. Please report the EC50 for auranofin. If possible, ideally include values for both models (HepG2.2.15.7 and primary human hepatocytes) .

2. Please assess whether auranofin affects host-protein secretion (e.g., ALB).

3. Please unify terminology (“HBsAg”).

The manuscript alternates between “HBs antigen” and “HBsAg.”

**Do you want your identity to be public for this peer review?** For information about this choice, including consent withdrawal, please see our Privacy Policy

Reviewer #1: No

Reviewer #2: No

---

## [Author Response · Author response to Decision Letter 1]

8 Dec 2025

Ms. Emily Chenette

Editor-in-Chief, PLOS One

Dr. Tatsuo Kanda

Academic Editor, PLOS One

Re: Manuscript ID: PONE-D-25-55761

Editors:

Thank you very much for your kind letter. We are grateful for your encouraging review of our manuscript (PONE-D-25-55761), titled “Auranofin, identified by FDA-approved drug library screening, inhibits HBs antigen secretion via lysosomal damage”. We have carefully considered your criticisms and comments and have revised our manuscript accordingly.

Point-by-point responses to the comments of the Reviewers are provided below. All changes are highlighted in red.

We hope that the revised manuscript is now suitable for publication in PLOS One.

Sincerely,

Hayato Hikita, M.D., Ph.D.

Department of Gastroenterology and Hepatology,

The University of Osaka Graduate School of Medicine,

2-2 Yamadaoka, Suita, Osaka 565-0871, Japan

Telephone number: +81-6-6879-3621

hikita@gh.med.osaka-u.ac.jp

Journal Requirements:

Reply

Thank you for pointing this out. We have thoroughly revised the manuscript and supporting files to ensure full compliance with the PLOS ONE style requirements, including file naming conventions. We have followed the formatting guidelines provided in the links and updated all relevant sections and files accordingly.

Reply

Thank you for bringing this to our attention. We apologize for the inconsistency between the ‘Funding Information’ and ‘Financial Disclosure’ sections. We have carefully reviewed the grant details and corrected the grant numbers in the ‘Funding Information’ section to ensure they accurately reflect the awards that supported this study. The revised sections are now consistent.

This study was funded by the Japan Agency for Medical Research and Development (AMED) under grant numbers JP25fk0310534 (K.M., H.H., T.K., and T. Taka.) and JP223fa627006 (Y.H., T. Taka., and H.S.), and by the Basis for Supporting Innovative Drug Discovery and Life Science Research (BINDS).

Reply

The BINDS provided the FDA-approved drug library used in this study. Aside from providing this material, the funders had no role in study design, data collection and analysis, decision to publish, or preparation of the manuscript.

4. Thank you for stating the following in the Competing Interests/Financial Disclosure section:

Tetsuo Takehara received research grants from Gilead Sciences, Inc, GSK plc. and Bristol Myers Squibb. Hayato Hikita, Yuki Tahata, and Tetsuo Takehara received lecture fees from Gilead Sciences.

We note that one or more of the authors are employed by commercial companies: Gilead Sciences, Inc, GSK plc. and Bristol Myers Squibb

Reply

Thank you for your comments. None of the authors are employed by Gilead Sciences, Inc., GSK plc., or Bristol Myers Squibb. These companies provided research grants or lecture fees to some authors, as disclosed in the Competing Interests/Financial Disclosure section, but no author receives salary support from these companies. Accordingly, we have amended the Funding Statement and Competing Interests Statement to clarify the sources of funding and the role of funders using the required wording. The updated Funding Statement and Competing Interests Statement is provided below. As requested, we will ensure that the Author Contributions section accurately reflects each author’s roles in the study.

Funding Statement:

This study was funded by the Japan Agency for Medical Research and Development (AMED) under grant number JP25fk0310534 (K.M., H.H., T.K., and T.Taka.) and JP223fa627006 (Y.H., T.Taka., and H.S.), and Basis for Supporting Innovative Drug Discovery and Life Science Research (BINDS). Tetsuo Takehara received research grants from Gilead Sciences, Inc., GSK plc., and Bristol Myers Squibb. Hayato Hikita, Yuki Tahata, and Tetsuo Takehara received lecture fees from Gilead Sciences. BINDS provided the FDA-approved drug library used in this study. The funders had no role in study design, data collection and analysis, decision to publish, or preparation of the manuscript.

Competing Interests Statement:

This study was funded by the Japan Agency for Medical Research and Development (AMED) under grant number JP25fk0310534 (K.M., H.H., T.K., and T.Taka.) and JP223fa627006 (Y.H., T.Taka., and H.S.), and Basis for Supporting Innovative Drug Discovery and Life Science Research (BINDS). Tetsuo Takehara received research grants from Gilead Sciences, Inc., GSK plc., and Bristol Myers Squibb. Hayato Hikita, Yuki Tahata, and Tetsuo Takehara received lecture fees from Gilead Sciences. None of the authors are employed by these commercial entities, and the authors declare that they have no other competing interests such as consultancy, patents, products in development, or marketed products. This does not alter our adherence to PLOS ONE policies on sharing data and materials.

Reply

The following statement has been added to the manuscript:

“Ethics statement

This study did not involve human participants, human data, or human tissue. Therefore, approval by an institutional review board or ethics committee and informed consent were not required. All experimental procedures involving animals were conducted in accordance with the institutional guidelines for animal care and were approved by the Animal Care Committee of the Central Institute for Experimental Medicine and Life Science.”

6. We note that your Data Availability Statement is currently as follows: All relevant data are within the manuscript and its Supporting Information files.

Reply

Thank you for your inquiry. We confirm that our submission contains all raw data required to replicate the results of this study. The numerical values underlying all graphs, summary statistics, and quantifications have now been compiled and uploaded as Supporting Information files. These files include the values behind means and standard deviations, the data used to generate graphs, and the quantified data points extracted from images. An updated Data Availability Statement has been added below.

Data Availability Statement:

All relevant data are within the manuscript and its Supporting Information files. The minimal data set underlying all figures and quantitative analyses, including numerical values used to generate graphs and data points extracted from images, has been provided as Supporting Information.

Reply

Thank you for your comment. The reviewers did not recommend any additional citations in their comments. We have carefully re-examined the reviewer reports to confirm that no citation requests were made. Therefore, no changes to the References section were required.

Review Comments

Reviewer #1:

The authors focused on auranofin as a potential novel therapeutic agent for HBV infection.

Auranofin failed to decrease intracellular HBsAg or HBV RNA. The authors showed that auranofin only reduced extracellular HBsAg release and increases the number of intracellular Galectin-3–positive vesicles in HBV-infected cells. Based on these findings, the authors concluded—and reflected in the title—that lysosomal damage is responsible for the reduced extracellular release of HBsAg. Since the major novelty of this study lies in the finding that auranofin is associated with reduced HBsAg release, it would be essential to elucidate the molecular mechanism underlying this effect. Without mechanistic evidence linking lysosomal damage directly to impaired HBsAg secretion, publication of this work would be difficult to justify.

Reply

We sincerely thank the reviewer for the insightful and constructive comments. We agree that elucidating the mechanistic link between auranofin-induced lysosomal damage and the reduction of extracellular HBsAg release is crucial for strengthening the novelty and significance of our study. In response to the reviewer’s comment, we have now conducted additional experiments to further clarify the underlying mechanism.

#1. While the increase in Galectin-3 and damaged lysosomes after auranofin treatment in HBV-infected HepaSH cells suggests lysosomal injury, this observation alone does not directly demonstrate that lysosomal damage contributes to the reduced extracellular release of HBsAg.The authors could perform rescue experiments using antioxidants or lysosomal membrane–stabilizing interventions to determine whether preventing lysosomal damage restores HBsAg secretion under Auranofin treatment.

Reply

- Thank you for the important suggestion. As the reviewer noted, we did not directly demonstrate that the increase in Galectin-3 contributes to the reduced extracellular release of HBsAg. Autophagy induced following lysosomal damage caused by B5, a thioredoxin reductase inhibitor similar to auranofin, has been reported to occur in a ROS-independent manner (Shao FY, et al. B5, a thioredoxin reductase inhibitor, induces apoptosis in human cervical cancer cells by suppressing the thioredoxin system, disrupting mitochondrion-dependent pathways and triggering autophagy. Oncotarget. 2015 Oct 13;6(31):30939-56. doi: 10.18632/oncotarget.5132.), and thus cannot be effectively rescued by antioxidants. We therefore examined the effect of overexpressing thioredoxin reductase—the known target inhibited by auranofin—on extracellular HBsAg levels.

In HepG2.2.15.7 cells, the thioredoxin reductases TXNRD1 and TXNRD2 were overexpressed. Compared with the control group, the group in which TXNRD1 was overexpressed did not have altered HBsAg levels in the culture supernatant, whereas the group in which TXNRD2 was overexpressed had increased extracellular HBsAg levels. We added these data to S4 Fig, and rewrote the Results section (p. 21 l 385).

Although TXNRD2 overexpression partially restored extracellular HBsAg release, we acknowledge that this approach may not directly demonstrate a causal relationship between lysosomal injury and impaired HBsAg secretion. To clarify this mechanistic link more conclusively, further studies aimed at directly modulating lysosomal integrity are needed, and this limitation has been noted in the Discussion (p. 23 l 440).

#2. The correlation between the degree of Galectin-3 expression or the number of Galectin-3–positive vesicles with the amount of extracellular HBsAg would strengthen the conclusion. Although no clear dose-dependency of auranofin was observed, this point is somewhat concerning, as it raises questions about the reproducibility and mechanism of the observed effects. A dose–response or time-course analysis showing that

---

## [Decision Letter · Decision Letter 1]

16 Dec 2025

Auranofin, identified by FDA-approved drug library screening, inhibits HBs antigen secretion via lysosomal damage

PONE-D-25-55761R1

Dear Dr. Hikita,

We’re pleased to inform you that your manuscript has been judged scientifically suitable for publication and will be formally accepted for publication once it meets all outstanding technical requirements.

Kind regards,

Tatsuo Kanda, M.D.; Ph.D.

Academic Editor

PLOS One

Additional Editor Comments (optional):

Reviewers' comments:

Reviewer's Responses to Questions

**Comments to the Author**

Reviewer #2: All comments have been addressed

Reviewer #3: All comments have been addressed

2. Is the manuscript technically sound, and do the data support the conclusions?

Reviewer #2: Yes

Reviewer #3: Yes

3. Has the statistical analysis been performed appropriately and rigorously?

Reviewer #2: Yes

Reviewer #3: Yes

4. Have the authors made all data underlying the findings in their manuscript fully available?

Reviewer #2: Yes

Reviewer #3: Yes

5. Is the manuscript presented in an intelligible fashion and written in standard English?

Reviewer #2: Yes

Reviewer #3: Yes

Reviewer #2: The authors have adequately addressed my concerns. Thank you for the opportunity to review this interesting study.

Reviewer #3: (No Response)

**Do you want your identity to be public for this peer review?** For information about this choice, including consent withdrawal, please see our Privacy Policy

Reviewer #2: No

Reviewer #3: No

---

## [Editor Report · Acceptance letter]

PONE-D-25-55761R1

PLOS One

Dear Dr. Hikita,

I'm pleased to inform you that your manuscript has been deemed suitable for publication in PLOS One. Congratulations! Your manuscript is now being handed over to our production team.

Kind regards,

on behalf of

Professor Tatsuo Kanda

Academic Editor

PLOS One